# Genetic diversity and evolutionary insights of Dali tea (*Camellia taliensis*) in the Lancang River Basin: Implications for tea breeding and resource conservation

Yanlan Tao[1,2☉‡], Lichao Huang[3☉‡], Hongyu Chen[2☉‡], Yiju Luo[2], Rong Tang[2], Faying Li[1,2*], Zengquan Lan[1,2*]

**1** College of Forestry, Southwest Forestry University, Yunnan, China, **2** Ancient Tea Tree Research Centre, Southwest Forestry University, Yunnan, China, **3** College of Horticulture and Landscape Architecture, Southwest Forestry University, Yunnan, China

☉ These authors contributed equally to this work.
‡ These authors share first authorship on this work.
* lifaying@swfu.edu.cn (FL); lanzengquan@tsinghua.org.cn (ZL)

## Abstract

Dali tea (*Camellia taliensis*), serving as a primitive wild species within the section Thea, represents a crucial genetic source for the domestication of Pu-erh tea (*C. sinensis* var. *assamica*) due to its strong stress tolerance and unique biochemical composition. It is of key value for the conservation of tea genetic resources and breeding innovation. Utilizing the SLAF-seq (Specific-Locus Amplified Fragment Sequencing) technique, this study systematically analyzed the genetic diversity and evolutionary relationships among five geographic populations (16 *C. taliensis* and 4 *C. sinensis* var. *assamica* accessions) within the Lancang River basin. Results revealed significant genetic differentiation among the *C. taliensis* populations. Pronounced genetic isolation was observed between the Lincang Daxueshan and Dali Nanjian populations. Localized gene introgression occurred between wild *C. taliensis* (Nanjian population) and *C. sinensis* var. *assamica*.The wild Lincang Daxueshan population formed a monophyletic clade at the base of the phylogenetic tree, exhibiting strong genetic isolation and high differentiation levels (Fst = 0.364) but low genetic diversity. In contrast, the cultivated population (Banna Germplasm Repository) displayed a mixed genetic background, with wild genetic components constituting only 50%−60%. The Lincang Daxueshan wild population showed a low minor allele frequency (MAF = 0.204) and a mild inbreeding coefficient (Fis = 0.09), indicating a potential risk of genetic erosion. Conversely, the Banna Germplasm Repository population exhibited the highest genetic diversity (Shannon Index = 0.318), highlighting the effectiveness of ex situ conservation and its potential as a vital gene donor for tea breeding. This study underscores the unique status of the upper Lancang River basin in Yunnan as a core conservation area for *C. taliensis* genetic diversity. We propose

**Data availability statement:** All relevant data are publicly available in the NCBI Sequence Read Archive (SRA) under accession number PRJNA1166700 (https://dataview.ncbi.nlm. nih.gov/object/PRJNA1166700? reviewer=vf1ede77gcukscli3ak5da3o2).

**Funding:** This study was supported by Yunnan Provincial Education Department fund project of China (2024J0670), Major science and technology project of Yunnan Province (202002AA100007), National Forestry and Grassland Administration sci-ence and technol-ogy project of China(2019130004-149).

**Competing interests:** The authors have declared that no competing interests exist.

strategies of "delineating priority zones for *in situ* conservation" and "facilitating inter-population germplasm exchange," providing a molecular basis for conserving wild tea resources and breeding for stress resistance. Employing high-density SNP markers, we obtained 5,182,931 loci with an average sequencing depth of 19.30x. This enabled quantification of gene flow between wild and cultivated populations (Nm = 0.18) and clarified the contribution of introgressive domestication to the genetic makeup of cultivated tea. These findings provide a theoretical foundation for under-standing interspecific interaction mechanisms in tea plant evolution and hold sig-nificant implications for promoting regional ecological conservation and biodiversity maintenance.

## Introduction

Dali tea (*Camellia taliensis (W. W. Sm.) Melch.*) [1,2], belonging to the section *Thea* of the genus *Camellia* L. (Theaceae), constitutes a vital component of tea germplasm resources. It harbors rich genetic diversity and holds irreplaceable value across multiple domains, including resource utilization, ecological conservation, cultural her-itage, and scientific research [3,4]. Yunnan, recognized as the global center of origin and diversity for tea plants, hosts the Lancang River basin. Its unique geographical barriers and heterogeneous habitats have shaped the distinctive distribution pattern and genetic architecture of *C. taliensis* [5,6]. Due to the primitive characteristics of its leaf biochemical composition and its reservoir of stress-resistance genes, *C. taliensis* serves as a crucial wild gene pool for deciphering the domestication mechanisms of cultivated tea (*C. sinensis* var. *assamica*) [7–9]. However, due to over-harvesting and habitat fragmentation, wild populations of *C. taliensis* are confronting a dual crisis of genetic diversity loss and adaptive decline. The reduction in their effective popula-tion size may exacerbate the risk of inbreeding depression [10]. Currently listed as Vulnerable (VU) on the IUCN Red List [11], they urgently require precise conservation strategies informed by genomic data [12]. Although previous studies have partially revealed the population genetic characteristics of *C. taliensis* using SSR markers [13] and chloroplast DNA fragments [7], research on closely related *Camellia* species suggests that traditional molecular markers may underestimate the complexity of genomic structural variation [14]. This limitation hinders the resolution of key scien-tific questions: (1) Whether the phylogeographic differentiation pattern of *C. talien-sis* within the Lancang River basin is coupled to the heterogeneity of its ecological corridors; (2) The spatiotemporal dynamics of gene flow between wild and cultivated populations and its contribution to domestication history; (3) The genomic distribution characteristics of candidate gene resources conferring adaptation to extreme envi-ronments. Recently, Huang et al. [14], employing reduced-representation genome sequencing, demonstrated that traditional molecular markers may underestimate the complexity of genetic structure in wild *Camellia*s pecies. In contrast, genome-wide SNP marker systems can effectively uncover adaptive evolutionary trajectories at microgeographic scales [15]. Specific-Locus Amplified Fragment Sequencing

(SLAF-seq) technology, based on targeted restriction enzyme site design, enables high-density SNP marker development even in the absence of a reference genome. Although its resolution is constrained by the distribution of enzyme recognition sites, its data throughput and cost-effectiveness are significantly superior to traditional sequencing approaches [16,17]. In practical applications for tea cultivation, this technology has successfully supported the identification of ancient tea tree resources [18] and the analysis of genetic structure in cultivated varieties [19]. However, its application to the conservation of wild tea germplasm resources remains subject to two major limitations: Firstly, existing studies still lack systematic analysis of the dynamic patterns of genomic introgression between wild and cultivated species; Secondly, a functional locus screening framework based on environmental adaptability has not yet been established, hindering the targeted utilization of wild gene resources in breeding programs. This study focuses on 16 wild *C. taliensis* populations from five geographic units within the Lancang River basin, alongside four cultivated *C. sinensis* var. *assamica* (Pu-er tea) accessions. Utilizing SLAF-seq technology, we aim to achieve the following objectives: (1) Construct a high-resolution genetic structure map of *C. taliensis* at the basin scale, revealing the geographic distribution pattern of its genetic diversity within the Lancang River basin; (2) Quantify the intensity of gene flow between wild and cultivated populations and characterize their genetic differentiation; (3) Provide preliminary genetic evidence to support *in situ* conservation strategies for *C. taliensis* germplasm resources. The findings will address the research gap concerning the interrelationships among baseline genetic diversity, population differentiation patterns, and conservation strategies for *C. taliensis* in the Lancang River basin. This will provide a molecular basis for the *in situ* conservation and sustainable utilization of tea genetic resources within this region.

## Results and analyses

### Restriction enzyme selection and library construction assessment

Using the *C. sinensis* genome as a reference, *in silico* restriction enzyme prediction was performed. The restriction enzyme HaeIII was ultimately selected, and fragments of 450–500 bp generated by digestion were defined as SLAF tags. Sequencing yielded 103.84 million (Mb) reads. Bioinformatic analysis developed 661,359 SLAF tags, of which 40,941 were polymorphic. Ultimately, 5,182,931 population-wide SNPs were obtained. Sequencing data quality parameters (Table 1) were validated by the Q30 value (average 93.32%), confirming data reliability suitable for subsequent analyses.

### Development of SLAF tags and SNP markers

The average sequencing depth of SLAF tags obtained from the 20 tea germplasm accessions was 19.30× (Table 2). The number of tags per sample ranged from 154,203–259,806. Among these, sample SHC1 (*C. taliensis* from Shanhua Village, Nanjian, Dali) exhibited both the highest number of tags and the highest sequencing depth. Sample SNP completeness ranged from 42.05% to 58.32%, with observed heterozygosity levels between 11.67% and 16.91% (mean 14.29%). These values indicate relatively high genomic heterozygosity, supporting the suitability of the data for population genetic relationship analysis. Chromosomal distribution analysis revealed significant enrichment of SLAF tags in specific genomic regions, notably within a 74 Mb window on chromosome 10, a 55 Mb window on chromosome 2, and a 5 Mb window on chromosome 13 (Fig 1). This pattern suggests the potential presence of species-specific genetic variations within these chromosomal segments.

### Analysis of population genetic structure

**Phylogenetic structure analysis.** The phylogenetic tree is employed to delineate taxonomic and evolutionary relationships among species [20]. Phylogenetic reconstruction based on 16 *C. taliensis* and 4 *C. sinensis* var. *assamica* accessions from five prefectures within the Lancang River basin revealed two distinct major clades (Fig 2). Six *C. taliensis* accessions from the Lincang Daxueshan population (DLC1, DLC3, QT1, QT2, DXS1, DXS2) clustered within

**Table 1. Summary statistics of SLAF-seq sequencing data for the 20 samples.**

| Sample ID | Total Reads | GC Percentage (%) | Q30  Percentage (%) |
|---|---|---|---|
| CMQ | 5,384,780 | 42.89 | 89.35 |
| CSHCS | 5,450,453 | 43.94 | 94.03 |
| DLC | 5,251,972 | 43.53 | 93.16 |
| DLC3 | 5,221,343 | 44.00 | 93.16 |
| DXS1 | 3,726,175 | 43.98 | 93.52 |
| DXS2 | 4,753,872 | 44.38 | 93.41 |
| DXS3 | 6,665,743 | 43.61 | 93.18 |
| DYK1 | 5,499,299 | 43.62 | 94.08 |
| DLC1 | 5,251,972 | 43.53 | 93.16 |
| DYK2 | 5,477,489 | 43.64 | 93.67 |
| LC | 4,677,307 | 43.74 | 93.42 |
| MJ | 4,584,794 | 43.60 | 93.23 |
| MNZ | 4,781,228 | 43.40 | 93.22 |
| QT1 | 5,971,369 | 43.33 | 93.62 |
| QT2 | 3,467,741 | 44.28 | 93.31 |
| SHC1 | 8,451,260 | 42.91 | 93.38 |
| SHC2 | 4,980,553 | 43.53 | 93.76 |
| SM1 | 4,755,811 | 43.41 | 93.30 |
| ZYP833 | 2,840,280 | 45.32 | 93.86 |
| ZYP865 | 6,593,089 | 44.00 | 94.44 |

**Table 2. Summary of SLAF tag and SNP information for individual samples.**

| Sample ID | Number of SLAF tags | Total sequencing depth | Average sequencing depth | SNP count | Integrity% | Heterozygosity% |
|---|---|---|---|---|---|---|
| CMQ | 255,123 | 4,142,450 | 16.2371 | 2,553,122 | 49.26% | 16.14% |
| CSHCS | 185,332 | 3,929,348 | 21.2017 | 2,570,681 | 49.60% | 13.30% |
| DLC1 | 183,866 | 3,765,022 | 20.4770 | 2,565,190 | 49.49% | 13.66% |
| DLC3 | 177,609 | 3,782,380 | 21.2961 | 2,519,602 | 48.61% | 13.52% |
| DXS1 | 158,313 | 2,686,200 | 16.9677 | 2,317,558 | 44.72% | 13.01% |
| DXS2 | 188,806 | 3,378,013 | 17.8914 | 2,668,455 | 51.49% | 13.69% |
| DXS3 | 184,036 | 4,746,623 | 25.7918 | 2,538,681 | 48.98% | 12.63% |
| DYK1 | 219,968 | 4,014,120 | 18.2487 | 2,769,315 | 53.43% | 16.05% |
| DYK2 | 244,446 | 4,539,570 | 18.5709 | 2,769,235 | 53.43% | 12.38% |
| LC | 175,697 | 3,379,297 | 19.2337 | 2,508,858 | 48.41% | 13.77% |
| MJ | 175,977 | 3,326,890 | 18.9053 | 2,497,629 | 48.19% | 13.67% |
| MNZ | 204,952 | 3,718,458 | 18.1431 | 2,563,165 | 49.45% | 12.32% |
| QT1 | 154,203 | 4,389,268 | 28.4642 | 2,179,390 | 42.05% | 11.67% |
| QT2 | 161,985 | 2,498,176 | 15.4223 | 2,373,291 | 45.79% | 13.26% |
| SHC1 | 259,806 | 6,993,904 | 26.9197 | 2,573,345 | 49.65% | 15.44% |
| SHC2 | 242,454 | 3,948,091 | 16.2839 | 2,894,433 | 55.85% | 15.39% |
| SM1 | 224,755 | 3,997,090 | 17.7842 | 2,585,283 | 49.88% | 11.91% |
| SX | 238,367 | 3,994,505 | 16.7578 | 2,995,491 | 57.80% | 16.44% |
| ZYP833 | 191,817 | 2,081,833 | 10.8532 | 2666295 | 51.44% | 15.20% |
| ZYP865 | 237,713 | 4,884,720 | 20.5488 | 3022446 | 58.32% | 16.91% |

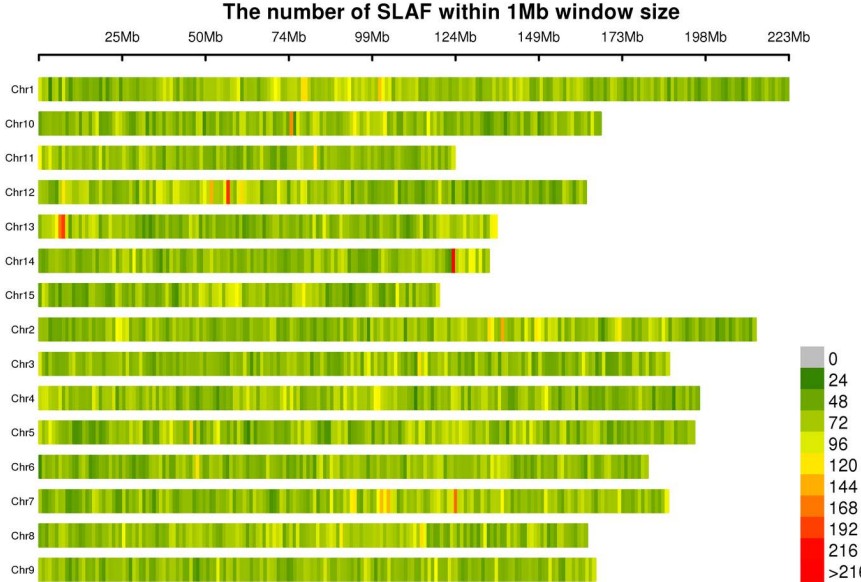

**Fig 1. Distribution of SLAF tags across chromosomes.** Note: Regions with increasing color intensity represent areas of concentrated SLAF tag distribution.

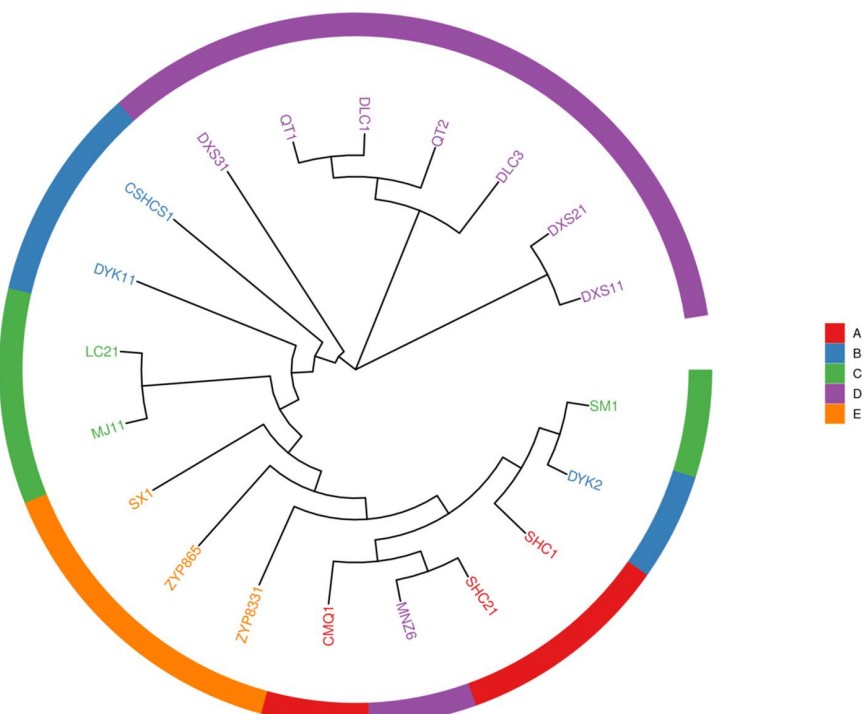

**Fig 2. Phylogenetic tree.** Note: Accessions are colored by population: A (red) – Dali Wuliangshan; B (blue) – Baoshan Changning; C (green) - Pu'er Ailaoshan; D (purple) – Lincang Daxueshan; E (orange) – Banna Germplasm Repository. Letters denote sample identifiers.

one clade, while the remaining 14 accessions (representing four other geographic populations) formed a separate major clade. Notably, the Lincang Daxueshan accessions (DXS1-DXS3, QT1-QT2) constituted a monophyletic clade with a homogeneous genetic background. In contrast, the cultivated accessions exhibited an admixed genetic background. The clustering pattern of the Pu'er Ailaoshan population accessions (MJ, LC) with germplasm from the Banna Repository (SX, ZYP833, ZYP866) showed a negative correlation with geographic distance (r = −0.32, *P* < 0.05), suggesting anthropogenic activities (e.g., germplasm introduction) may have disrupted natural dispersal patterns [21]. The strong concordance between geographic distribution and genetic clustering indicates closer kinship among individuals from the same locality. A weak genetic association was observed solely between *C. taliensis* from Dali Nanjian and *C. sinensis* var. *assamica* from Lincang Daxueshan, indicating potential localized interspecific gene flow.

### Population structure and principal component analysis (PCA)

Population structure analysis is a widely used clustering method that quantifies the number of ancestral populations and infers the ancestry proportion of each sample [22]. Population structure analysis revealed that the cross-validation error rate was minimized at K = 2 (Fig 3). Accordingly, the 20 accessions were clearly partitioned into two distinct and cohesive genetic clusters (Fig 4). Cluster I comprised eight wild *C. taliensis* accessions from Lincang Daxueshan and Baoshan Changning (CSHCS, DLC1, DLC3, DXS1, DXS2, DXS3, QT1, QT2). This cluster exhibited a homogeneous genetic background, deriving entirely from a single ancestral population (Ancestor 1, represented by one color). Cluster II consisted of six accessions, including cultivated *C. sinensis* var. *assamica* and cultivated *C. taliensis* (CMQ, DYK2, MNZ, SHC1, SHC2, SM1). This cluster carried genetic information primarily from a second ancestral population (Ancestor 2, blue), indicating pronounced genetic divergence between wild *C. taliensis* and *C. sinensis* var. *assamica*. The remaining six accessions (MJ, LC, DYK1, ZYP865, ZYP833, SX) showed admixed ancestry. This suggests the introgression of wild genetic resources into some cultivated *C. taliensis* accessions, indicating unidirectional gene flow from wild to cultivated species. These individuals likely represent hybrids derived from the two ancestral subpopulations. However, accessions

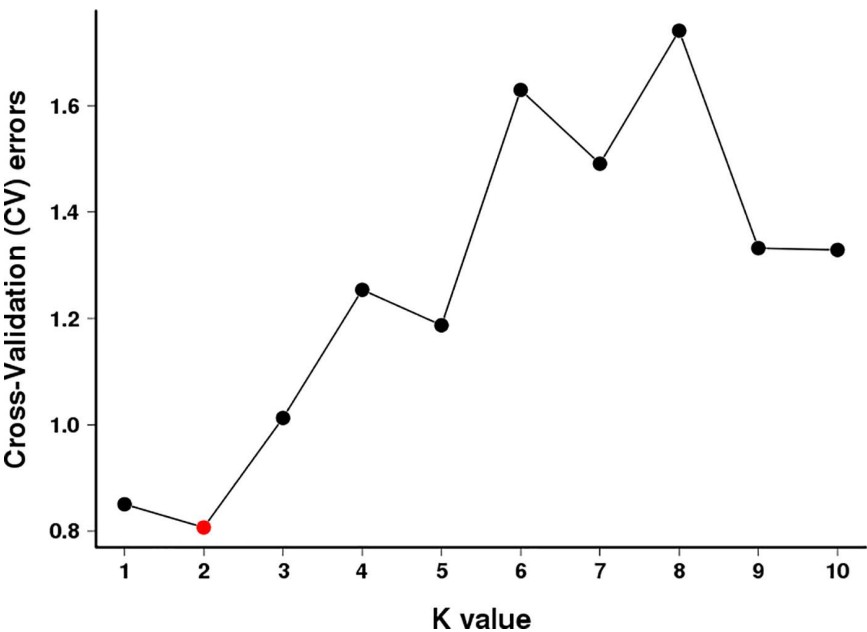

**Fig 3. Cross-validation error rate for different K values.** Note: The x-axis indicates the K-value (ranging from 1 to 10), while the y-axis displays the values of cross-validation error.

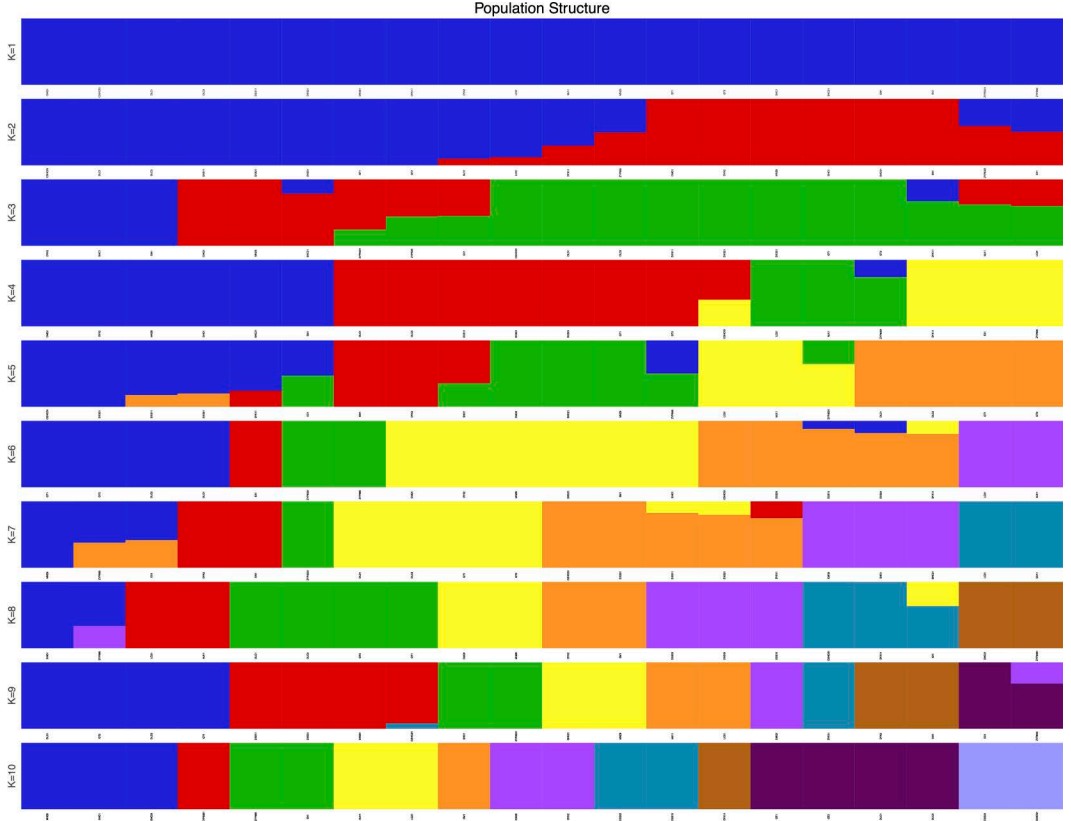

**Fig 4. Sample clustering results corresponding to different K value.** Note: The horizontal axis delineates 20 tea samples arranged sequentially, while the vertical axis indicates the number of subgroups denoted by K values (K = 1–10). Distinct colors are utilized to signify subgroups characterized by varying gene frequencies across the 20 tea plant samples, with tea plants within the same subgroup exhibiting close genetic relationships. The color assigned to each sample, along with its proportional representation, reflects the subgroup affiliation of the sample and the relative contribution of genetic material from that subgroup.

MJ, LC, and DYK1 retained ≥70% of the wild *C. taliensis* genetic background, classifying them as transitional types resulting from wild germplasm introgression. Cultivated *C. taliensis* accessions from the Banna Germplasm Repository (ZYP865, ZYP833, SX) retained relatively lower proportions of the wild *C. taliensis* genetic background, with ancestry contributions of 60%, 50%, and 50%, respectively. The wild Lincang Daxueshan population showed no further substructure at K-values ≥2, indicating its distinct genetic background with minimal or no gene flow from other species, supporting its genetic isolation. In contrast, populations from Baoshan Changning, Dali Nanjian, Pu'er Ailaoshan, and the Banna Germplasm Repository exhibited significant substructure, reflecting complex patterns of both interspecific and intraspecific gene flow. Principal Component Analysis (PCA; Fig 5) further corroborated the population structure results. *C. taliensis* and *C. sinensis* var. *assamica* accessions showed significant spatial separation along the principal components, confirming their distant kinship and substantial genetic divergence. With the exception of some overlap between *C. sinensis* var. *assamica* accessions from Baoshan Changning and Pu'er Ailaoshan, accessions from Lincang Daxueshan and Dali Nanjian were dispersed across the PCA plot. The distribution of *C. taliensis* accessions correlated strongly with their geographic origin. Accessions from Lincang Daxueshan and Baoshan Changning clustered tightly, while others showed genetic differentiation consistent with geographic isolation, aligning with conclusions from the phylogenetic tree and population structure analyses.

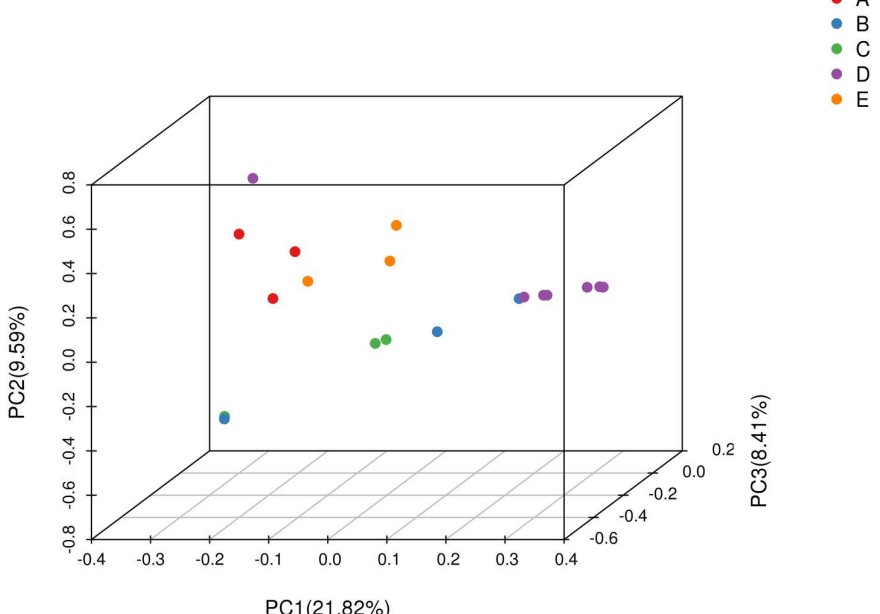

**Fig 5. Principal component analysis (PCA) plot.** Note: Each point represents one sample. Points are colored by population: A (red) – Dali Wuliang-shan; B (blue) – Baoshan Changning; C (green) - Pu'er Ailaoshan; D (purple) – Lincang Daxueshan; E (orange) – Banna Germplasm Repository.

## Analysis of genetic diversity

**Genetic diversity parameter analysis.** Studies of genetic diversity can elucidate the evolutionary history of species or populations (including their time and mode of origin) and provide critical insights for assessing their evolutionary potential and future trajectories [23]. Genetic diversity parameters for the five *C. taliensis* populations (Table 3) showed moderate levels for both the expected number of alleles (1.294–1.365) and the observed number of alleles (1.434–1.651). The mean expected heterozygosity (He) and observed heterozygosity (Ho) were 0.213 and 0.168, respectively. Nei's diversity index ranged from 0.202 to 0.265 (mean 0.233), and the Shannon-Wiener index ranged from 0.251 to 0.318 (mean 0.290). The *C. taliensis* population from the Banna Germplasm Repository exhibited the highest values for expected number of alleles (1.365), expected heterozygosity (0.214), observed heterozygosity (0.245), Nei's diversity index (0.265), and Shannon-Wiener index (0.318), indicating rich genetic diversity and potentially high environmental adaptability. The Baoshan Changning population ranked second (Shannon index = 0.298). In contrast, the Dali Nanjian, Pu'er Ailaoshan, and Lincang Daxueshan populations displayed relatively lower genetic diversity levels. The Lincang Daxueshan *C. taliensis* population showed significant minor allele frequency (MAF) depletion (MAF = 0.204, compared to 0.288–0.303 in other populations). Its observed heterozygosity (Ho = 0.172) and expected heterozygosity (He = 0.189) were lower than those of the cultivated Banna Repository population (Ho = 0.245, He = 0.214). The observed Ho < He in Lincang Daxueshan may reflect inbreeding or genetic drift [24]. While He exceeded Ho in most populations, the cultivated Banna Repository population exhibited the opposite pattern (Ho > He), suggesting it may have undergone unique genetic dynamics, such as artificial selection or significant gene flow. Furthermore, the Polymorphism Information Content (PIC) value reflects the ability of genetic markers to detect genetic variation. The PIC values across all five populations were low (0.135–0.171), indicating that the current markers have limited power for detecting genetic variation within these *C. taliensis* populations. Developing markers with higher polymorphism is necessary to enhance detection sensitivity. Overall, the analyzed *C. taliensis* germplasm resources exhibited moderate genetic diversity, likely associated with habitat heterogeneity and reproductive isolation within their contemporary distribution range.

**Table 3. Summary of genetic diversity parameters for the different populations.**

| Statistical values of genetic diversity | Population | | | | |
|---|---|---|---|---|---|
| | DL | BS | PE | LC | BN |
| Average_MAF | 0.302 | 0.303 | 0.299 | 0.204 | 0.288 |
| Expected_allele_number | 1.294 | 1.351 | 1.336 | 1.302 | 1.365 |
| Expected_heterozygous_number | 0.17 | 0.202 | 0.194 | 0.189 | 0.214 |
| Nei_diversity_index | 0.21 | 0.247 | 0.239 | 0.202 | 0.265 |
| Observed_allele_number | 1.434 | 1.514 | 1.496 | 1.651 | 1.562 |
| Observed_heterozygous_number | 0.166 | 0.174 | 0.15 | 0.172 | 0.245 |
| Polymorphism_information_content | 0.135 | 0.160 | 0.154 | 0.155 | 0.171 |
| Shannon_Wiener_index | 0.251 | 0.298 | 0.287 | 0.296 | 0.318 |

Note: DL: Dali Nanjian population; BS: Baoshan Changning population; PE: Pu'er Ailaoshan population; LC: Lincang Daxueshan population; BN: Banna Germplasm Repository population

**Population differentiation and conservation priority.** Based on Wright's [25] classification of the fixation index (Fst): $0 < Fst < 0.05$ indicates low genetic differentiation; $0.05 \leq Fst < 0.15$ indicates moderate genetic differentiation; $0.15 \leq Fst < 0.25$ indicates high genetic differentiation; and $Fst \geq 0.25$ indicates very high genetic differentiation. Pairwise Fst values (Table 4) revealed high differentiation between Dali Nanjian (A) and Baoshan Changning (B), Pu'er Ailaoshan (C), and Banna Repository (E), and very high differentiation with Lincang Daxueshan (D). Baoshan Changning (B) showed moderate differentiation with Pu'er Ailaoshan (C), Lincang Daxueshan (D), and Banna Repository (E). Pu'er Ailaoshan (C) exhibited moderate differentiation with Banna Repository (E) and high differentiation with Lincang Daxueshan (D). Banna Repository (E) showed high differentiation with Lincang Daxueshan (D). In summary, all pairwise comparisons indicated moderate to very high differentiation levels. Genetic differentiation between Lincang Daxueshan (D) and all other populations was exceptionally strong. This differentiation pattern correlates strongly with geographical isolation and habitat divergence. The isolated geography of Lincang Daxueshan likely impeded gene flow, confirming its unique status as a potential genetic refuge for *C. taliensis* [26]. Gene flow estimates indicated weak gene flow (Nm = 0.18) between Dali Nanjian (A) and Pu'er Ailaoshan (C), potentially reflecting unidirectional introgression from cultivated to wild types. Stronger gene flow (Nm = 1.30) occurred between Lincang Daxueshan (D) and Banna Repository (E), further evidencing genetic exchange facilitated by human intervention. The co-occurrence of low genetic diversity and high differentiation signals a risk of genetic erosion. Establishing ecological corridors to facilitate gene flow among populations is critical to enhance adaptive potential [27].

**Table 4. Pairwise Fst values among populations.**

| | A | B | C | D | E |
|---|---|---|---|---|---|
| A | | | | | |
| B | 0.201 | | | | |
| C | 0.217 | 0.139 | | | |
| D | 0.364 | 0.144 | 0.190 | | |
| E | 0.190 | 0.127 | 0.129 | 0.161 | |

**Note:** A: Dali Nanjian population; B: Baoshan Changning population; C: Pu'er Ailaoshan population; D: Lincang Daxueshan population; E: Banna Germplasm Repository population.

## Discussion

### Analysis of genetic diversity in *C. taliensis* from the Lancang River Basin and implications for conservation and breeding

Genetic diversity within germplasm resources forms the foundation for their utilization and exploitation in genetic breeding. Assessing the level of this diversity is crucial for identifying superior resources, selecting elite germplasm, and facilitating germplasm innovation in breeding programs [28]. The Polymorphic Information Content (PIC) reflects the level of diversity exhibited by a locus within a population [29,30]. According to standard interpretation [31], PIC > 0.5 indicates a highly polymorphic locus and high population genetic diversity; 0.25 < PIC < 0.5 indicates a moderately polymorphic locus and moderate genetic diversity; PIC < 0.25 indicates low genetic diversity. This study revealed a significantly lower average PIC value (PIC = 0.155) for *C. taliensis* populations in the Lancang River basin compared to previous studies. Mao et al. [32], investigating genetic diversity in wild and cultivated *C. taliensis* from three different populations, reported PIC values ranging from 0.041 to 0.877, with a mean of 0.491 (0.25 < PIC < 0.5), indicating moderate genetic diversity levels. Huang et al. [33], in a molecular identification study of 26 tea plant varieties under new plant variety protection application and 13 similar varieties, reported a PIC of 0.51 (PIC > 0.5) for these 39 tea germplasms, signifying high genetic diversity. Liu et al. [34] and Zhou et al. [35], analyzing tea plant resources from various regions and populations in Yunnan Province, documented PIC values reaching up to 0.527. However, our findings align with those of Ji et al. [36], who also studied Yunnan *C. taliensis*. This relatively low PIC likely stems from the natural attributes of *C. taliensis* as an endemic species with a narrow distribution. Habitat fragmentation restricts gene flow, and anthropogenic disturbances further exacerbate genetic isolation [36], reflecting the endangered status of this species. Notably, the mean observed heterozygosity (Ho = 0.181) across the five geographically distinct populations was lower than the mean expected heterozygosity (He = 0.194). This suggests prevalent inbreeding or genetic drift within the Lancang River basin *C. taliensis* populations. The Banna Germplasm Repository population exhibited the highest genetic diversity (Shannon index = 0.318), validating the effectiveness of *ex situ* conservation strategies [37]. Serving as a "genetic diversity hotspot," this population offers valuable material for mining stress-resistance genes [38]. Conversely, the low minor allele frequency (MAF = 0.204) and mild inbreeding coefficient (Fis = 0.09) in the wild Lincang Daxueshan population signal a risk of genetic diversity erosion. Urgent measures, including delineating core areas for *in situ* protection and implementing assisted migration, are needed to maintain its evolutionary potential [39]. Our study also detected gene introgression (Nm = 0.18) between wild and cultivated populations. Populations such as Baoshan Changning and Pu'er Ailaoshan retained 70%−90% wild genetic components, while the Banna Repository population displayed an admixed background. This provides direct evidence supporting the hypothesis that cultivated *C. taliensis* may have originated through introgressive domestication from wild germplasm [40]. Prioritizing the use of these transitional types in breeding programs is recommended to enhance adaptability. Furthermore, the strong genetic differentiation (Fst = 0.364) between the Lincang Daxueshan and Dali Nanjian populations suggests they may have experienced divergent natural selection pressures or prolonged geographical isolation. Subsequent genome scans could identify adaptive genes, laying the groundwork for marker-assisted breeding.

### Genetic structure analysis of *C. taliensis* in the Lancang River Basin and resource management strategies

Population structure analysis is indispensable in genetic diversity studies for elucidating genetic relatedness among germplasm resources and tracing the origins of specific genetic loci. Phylogenetic trees, a widely adopted approach, classify distinct germplasm accessions based on genetic proximity, thereby delineating kinship relationships and evolutionary trajectories. While Principal Component Analysis (PCA) enables intuitive visualization of genetic structure, it often lacks quantitative precision for determining optimal population subdivisions [41]. Single Nucleotide Polymorphism (SNP) markers are extensively employed in genetic map construction, genome-wide association studies (GWAS), and quantitative

trait analysis due to their stability and abundance of genetic variation [42,43]. SNP-based population structure analysis revealed that the Lincang Daxueshan wild population forms a monophyletic clade occupying the basal position in the phylogenetic tree. This ancestral group subsequently diverged to generate the other four *C. taliensis* populations and the *C. sinensis* var. *assamica* group. This finding indicates that the Lincang Daxueshan population retains more ancestral genetic characteristics than other groups and underscores its significance in the historical domestication and utilization of *C. taliensis*. The observation that *C. taliensis* exhibits more primitive evolutionary traits than *C. sinensis* var. *assamica* aligns with morphological evolutionary pathways proposed for *Sect. Thea* species by Chen et al. [44]. This further suggests the region served as a glacial refugium for *C. taliensis*, with its genetic distinctiveness providing critical insights into tea plant origins [45]. Nevertheless, this population's low genetic diversity and high differentiation imply constrained ecological adaptability. Strategic germplasm exchange across populations is thus essential to broaden its genetic base and enhance climate resilience [46]. Notably, the admixed clustering pattern between the Dali Nanjian population and *C. sinensis* var. *assamica* accessions supports the hypothesis by Li et al. [47] of *C. taliensis* involvement in Pu-erh tea domestication. Transcriptomic evidence from the Kunming Institute of Botany [2] corroborates this evolutionary relationship. We propose prioritizing the Nanjian population as a donor for interspecific hybridization to introgress wild-adaptive alleles. Furthermore, subpopulation differentiation within cultivated accessions (e.g., Xishuangbanna ZYP series) reflects anthropogenic reshaping of genetic architecture. Future breeding programs should emphasize geo-adaptive matching to prevent genetic homogenization from indiscriminate introduction [48]. Although SLAF-seq generated high-density SNP data, its genomic coverage bias toward gene-rich regions [18] likely underestimated diversity in repetitive and regulatory sequences. Additionally, rare alleles from peripheral populations were inadequately captured despite sampling major distribution zones. Subsequent research will employ whole-genome resequencing to detect structural variants and domestication signals, integrate phenomic data for genotype-phenotype association networks, and implement longitudinal monitoring to evaluate conservation efficacy.

## Materials and methods

### Materials

Plant materials comprised 16 accessions of *C. taliensis* and 4 accessions of *C. sinensis* var. *assamica* (Table 5) collected from five prefectures/cities within the Lancang River Basin. Sampling locations were selected based on the following criteria: 1) coverage of the geographic gradient from upper to lower reaches of the Lancang River Basin (elevation range: 1000–2700 m); 2) inclusion of both core distribution areas and marginal populations of *C. taliensis*; 3) use of *C. sinensis* var. *assamica* as a cultivated relative species for comparative genetic differentiation analysis. Sample size determination considered population distribution density within the basin and prior population genetics experience, ensuring a minimum of three biological replicates per geographic unit. Approximately 50 g of fresh, young leaves from the current year's growth were flash-frozen in liquid nitrogen and subsequently stored at −80°C for future use. Four accessions of *C. sinensis* var. *assamica* (SHC1, DYK2, SM1, MNZ) were collected with prior permissions granted by the respective County Agriculture and Rural Affairs Bureaus and Village Committees in Nanjian, Changning, Jingdong, and Shuangjiang counties.Sampling of cultivated *C. taliensis* accessions (ZYP833, ZYP865, SX, CMQ, SHC2) was authorized by the Tea Germplasm Resource Garden, Xishuangbanna Dai Autonomous Prefecture Academy of Agricultural Sciences, and the Nanjian Yi Autonomous County Agriculture and Rural Affairs Bureau.Wild *C. taliensis* accessions CSHCS and DYK1 were collected under permit from the Changning County Forestry and Grassland Bureau.The remaining nine wild *C. taliensis* accessions were sampled with permissions obtained from the Management Bureaus of the Yunnan Ailaoshan National Nature Reserve and the Yunnan Lincang Daxueshan National Nature Reserve.All tissue sampling was conducted under the supervision of local agricultural and forestry field specialists. Samples were used exclusively for scientific research purposes. The non-invasive sampling methods employed in this study had no detectable impact on the natural growth of the Camellia plants.

**Table 5. Tea sample table information.**

| Serial number | Number | Name | Species | Population | Source | Altitude |
|---|---|---|---|---|---|---|
| 1 | CMQ | Chimaqing Tea Tree | *C. taliensis* | Wuliangshan Mountain in Nanjian | Cultivated | 2,229 |
| 2 | SHC1 | Shanhua Village 1 | *C. sinensis* var. *assamica* | Wuliangshan Mountain in Nanjian | Cultivated | 2,221 |
| 3 | SHC2 | Shanhua Village 2 | *C. taliensis* | Wuliangshan Mountain in Nanjian | Cultivated | 2,226 |
| 4 | CSHCS | Chashan River tea tree | *C. taliensis* | Changning in Baoshan | Undomesticated | 2,348 |
| 5 | DYK1 | Tai Ya Kou 1 | *C. taliensis* | Changning in Baoshan | Undomesticated | 2,054 |
| 6 | DYK2 | Tai Ya Kou 2 | *C. sinensis* var. *assamica* | Changning in Baoshan | Cultivated | 2,085 |
| 7 | MJ | Mojiang tea tree | *C. taliensis* | Ailao Mountain in Pu'er | Undomesticated | 1,910 |
| 8 | LC | Lancang tea tree | *C. taliensis* | Ailao Mountain in Pu'er | Undomesticated | 2,021 |
| 9 | SM1 | Simao 1 | *C. sinensis* var. *assamica* | Ailao Mountain in Pu'er | Cultivated | 1,409 |
| 10 | DLC1 | Dali Tea 1 | *C. taliensis* | Daxueshan Mountain in Lincang | Undomesticated | 2,387 |
| 11 | DLC3 | Dali Tea 3 | *C. taliensis* | Daxueshan Mountain in Lincang | Undomesticated | 2,622 |
| 12 | QT1 | Dali Tea 1 | *C. taliensis* | Daxueshan Mountain in Lincang | Undomesticated | 2,100 |
| 13 | QT2 | Dali Tea 2 | *C. taliensis* | Daxueshan Mountain in Lincang | Undomesticated | 2,059 |
| 14 | DXS1 | Daxueshan 1 | *C. taliensis* | Daxueshan Mountain in Lincang | Undomesticated | 2,648 |
| 15 | DXS2 | Daxueshan 2 | *C. taliensis* | Daxueshan Mountain in Lincang | Undomesticated | 2,690 |
| 16 | DXS3 | Daxueshan 3 | *C. taliensis* | Daxueshan Mountain in Lincang | Undomesticated | 2,659 |
| 17 | MNZ | Meinanzi tea tree | *C. sinensis* var. *assamica* | Daxueshan Mountain in Lincang | Cultivated | 2,580 |
| 18 | ZYP833 | Resource Garden 833 | *C. taliensis* | Resource Garden in Banna | Cultivated | 1,167 |
| 19 | ZYP865 | Resource Garden 865 | *C. taliensis* | Resource Garden in Banna | Cultivated | 1,166 |
| 20 | SX | Shuixiecha tea tree | *C. taliensis* | Resource Garden in Banna | Cultivated | 1,166 |

## DNA extraction and quality control

Genomic DNA was extracted from all 20 tea plant accessions using the Broad-Spectrum Plant DNA Extraction Kit (Bio-mad, Beijing, China). Extracted DNA was assessed for quality, concentration, and purity using a gel imaging analysis system and a UV spectrophotometer (Thermo Fisher Scientific, USA). All samples exhibited OD260/280 ratios between 1.8 and 2.0, with concentrations ≥ 30 ng/μL. Qualified DNA aliquots were stored at −20°C. Three technical replicates per sample were prepared for subsequent quality control procedures.

## Enzymatic library construction

As the *C. taliensis* genome sequence is not publicly available, restriction enzyme digestion sites were predicted using the *C. sinensis* reference genome (available at http://tpia.teaplant.org/download.html#), based on the estimated genome size and GC content of *C. taliensis*. SLAF-seq (Specific-Locus Amplified Fragment Sequencing) was selected over GBS (Genotyping-by-Sequencing) or RAD-seq (Restriction-site Associated DNA Sequencing) based on: 1) the high polymorphism detection rate (>82%) of the *HaeIII* + *MseI* enzyme combination within the genus *Camellia*; 2) the flexibility to adjust tag density (target number of SLAF tags: 50,000); 3) suitability for population evolutionary analysis in complex genomes. Target fragments of 400–450 bp were predicted using SLAF_predict software. Following A-tailing of the 3' ends, sequencing adapters were ligated [49]. The optimal number of PCR amplification cycles, determined by gradient optimization, was 12. Libraries passing quality control were subjected to paired-end sequencing (2 × 50 bp) on the Illumina HiSeq 2000 platform.

## SLAF tag acquisition and SNP marker development

Paired-end reads were clustered into Specific Locus Amplified Fragment (SLAF) tags based on sequence similarity (≥95%) and positional consistency. To validate genotyping reliability, 5% of samples were randomly selected

for replicate library construction and sequencing, achieving a genotype concordance rate of 98.5%. Polymorphic SLAF tags were filtered using the following criteria: 1) minor allele frequency (MAF) ≥ 0.05 across the population; 2) sequencing depth ≥ 4× per sample; 3) genotype call rate ≥ 85%. As tea plants are diploid, a single locus can harbor up to four genotypes. Therefore, SLAF tags containing two, three, or four alleles were classified as polymorphic. SNP markers underwent stringent quality control: 1) loci with abnormal heterozygosity (>30%) were removed; 2) loci significantly deviating from Hardy-Weinberg equilibrium (P < 0.001) were excluded; 3) markers in strong linkage disequilibrium ($r^2 > 0.8$) were discarded. A high-confidence SNP dataset was ultimately obtained for downstream analyses.

## Data analysis

Reads from all 20 samples were clustered based on sequence similarity, with groups of reads sharing high similarity defined as individual SLAF tags. SLAF tags exhibiting sequence variations among different samples were identified as polymorphic SLAF tags. The most frequent sequence variant within each SLAF tag served as the reference sequence. Sequencing reads were aligned to the reference genome using BWA [50]. SNP calling was performed independently using GATK [51] and SAMtools [52]. The intersection of SNPs identified by both methods constituted the final high-confidence SNP marker dataset. Population-specific SNP loci were subsequently identified through comparative analysis. Phylogenetic analysis was conducted using MEGA X [53]. A neighbor-joining tree was constructed under the Kimura 2-parameter model to infer evolutionary relationships among samples. Population structure was assessed using ADMIXTURE [54] (cross-validation replicates = 10). Principal Component Analysis (PCA) was performed using EIGENSOFT [55]. The reliability of all analytical pipelines was verified using positive controls.

## Conclusions

This study employed SLAF-seq markers to develop SNP loci and analyze genetic diversity across 20 tea germplasm accessions from five geographic populations in the Lancang River Basin. Our results revealed significant geographic-genetic structuring within *C. taliensis* populations. Genetic introgression was detected between wild and cultivated groups. However, core wild populations (e.g., Lincang Daxueshan) exhibited latent risks associated with low genetic diversity. Low genetic diversity may compromise adaptive capacity to environmental change, necessitating prioritized establishment of *in situ* conservation sites to preserve gene pool integrity. Conversely, the high-diversity Xishuangbanna Germplasm Repository population represents a valuable genetic donor for tea breeding. Moderate to high genetic differentiation among populations underscores the need to consider genetic background differences during cross-regional germplasm introduction, preventing genetic homogenization through indiscriminate hybridization. This study provides a molecular theoretical framework for precision conservation of tea resources and parental selection in breeding. It particularly highlights the role of geographic isolation in shaping genetic structure, while expanding current understanding of interspecific gene flow and adaptive evolution in tea plant phylogenetics.

## Acknowledgments

We express our profound gratitude to the following institutions for their indispensable assistance during fieldwork: Nanjian Yi Autonomous County Bureau of Agriculture and Rural Affairs, Changning County Bureau of Agriculture and Rural Affairs and Changning County Forestry and Grassland Bureau, Jingdong Yi Autonomous County Bureau of Agriculture and Rural Affairs, Shuangjiang Lahu-Va-Blang-Dai Autonomous County Bureau of Agriculture and Rural Affairs, Village Committees of Nanjian, Changning, Jingdong, and Shuangjiang Counties, Tea Germplasm Resource Garden, Xishuangbanna Dai Autonomous Prefecture Academy of Agricultural Sciences, Administration Bureau of Yunnan Ailao Mountains National Nature Reserve, Administration Bureau of Yunnan Lincang Daxueshan National Nature Reserve.

## Author contributions

**Investigation:** Yanlan Tao, Lichao Huang, Hongyu Chen, Yiju Luo, Rong Tang.

**Project administration:** Faying Li, Zengquan Lan.

**Resources:** Zengquan Lan.

**Writing – original draft:** Yanlan Tao, Hongyu Chen.

**Writing – review & editing:** Yanlan Tao, Lichao Huang, Faying Li, Zengquan Lan.

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
