## [Decision Letter · Decision Letter 0]

Dear Dr. Tao,

Thank you for submitting your manuscript to PLOS ONE. After careful consideration, we feel that it has merit but does not fully meet PLOS ONE’s publication criteria as it currently stands. Therefore, we invite you to submit a revised version of the manuscript that addresses the points raised during the review process.

We look forward to receiving your revised manuscript.

Kind regards,

Ghulam Yaseen, Ph.D.

Academic Editor

PLOS ONE

Journal Requirements:

 “This study was supported by Yunnan Provincial Education Department fund project of China (2024J0670), Major science and technology project of Yunnan Province (202002AA100007), National Forestry and Grassland Administration sci-ence and technology project of China(2019130004-149).”        

5.  We note that Figure 6 in your submission contain [map/satellite] images which may be copyrighted. All PLOS content is published under the Creative Commons Attribution License (CC BY 4.0), which means that the manuscript, images, and Supporting Information files will be freely available online, and any third party is permitted to access, download, copy, distribute, and use these materials in any way, even commercially, with proper attribution. For these reasons, we cannot publish previously copyrighted maps or satellite images created using proprietary data, such as Google software (Google Maps, Street View, and Earth). For more information, see our copyright guidelines: http://journals.plos.org/plosone/s/licenses-and-copyright .

    a. You may seek permission from the original copyright holder of Figure 6 to publish the content specifically under the CC BY 4.0 license. 

Reviewers' comments:

Reviewer's Responses to Questions

**Comments to the Author**

1. Is the manuscript technically sound, and do the data support the conclusions?

Reviewer #1: Yes

Reviewer #2: Yes

2. Has the statistical analysis been performed appropriately and rigorously?

Reviewer #1: Yes

Reviewer #2: Yes

3. Have the authors made all data underlying the findings in their manuscript fully available?

Reviewer #1: Yes

Reviewer #2: No

4. Is the manuscript presented in an intelligible fashion and written in standard English?

Reviewer #1: Yes

Reviewer #2: Yes

Reviewer #1: Dear Authors

The abstract is overly detailed in certain aspects, such as numerical specifics of GC content and Q30, which are unnecessary for a general summary. However, it does not clearly convey the practical implications of the findings, nor does it highlight the novelty of the study. Also, the abstract lacks a clear statement of what differentiates this study from prior work on genetic diversity in tea plants.

Introduction

While the introduction provides a comprehensive background on Dali tea, it lacks a strong focus on the specific research gap addressed. The discussion of previous studies could be more targeted to emphasize what this study aims to contribute. The introduction could better highlight what aspects of Dali tea’s genetic structure or diversity this study uniquely addresses.

Materials and methods

Some methodological details are unclear, such as the rationale behind sample selection and the choice of SLAF-seq over other genotyping methods. The lack of information on statistical validation also raises questions about data reliability.

Results

Results are presented with an excess of technical data, such as base percentages, which detracts from clarity. The emphasis on minor details makes it difficult to identify key findings. There’s also a lack of critical interpretation; for example, the impact of low genetic diversity on conservation isn't discussed in detail. The results on genetic diversity and population structure add some new insights but lack a clear demonstration of unique findings within the broader field of tea genetics research.

Discussion

The discussion does not sufficiently relate the study’s results to conservation practices or breeding implications, which are only mentioned briefly. There is a lack of critical analysis of the limitations of the study’s data.

Reviewer #2: The Manuscript entitled "Genetic diversity and evolutionary insights of Dali Tea (Camellia taliensis) in the

Lancang River Basin: implications for tea breeding and resource conservation" is a well written and informative study that Dali tea species exhibit a higher presence of ancestral genetic traits than Pu'er tea species, indicating that Dali tea

might have contributed to Pu'er tea domestication.

Although the paper is very informative but the figures are not of publication standard, they need to be modified more and should have more explanatory notes. Figure legends should be more elaborative depicting the clear idea or concept of the pictorial representation.

Lastly, in some section the English can be better so we recommend to check the whole manuscript with professional English check.

**Do you want your identity to be public for this peer review?** For information about this choice, including consent withdrawal, please see our Privacy Policy

Reviewer #1: No

Reviewer #2: **Yes: ** Udayan Bhattacharya

---

## [Author Response · Author response to Decision Letter 1]

16 Jun 2025

We sincerely appreciate the opportunity to revise our manuscript and thank the reviewers for their constructive feedback. We have thoroughly addressed all comments point-by-point, with major revisions highlighted in yellow in the manuscript. Key improvements include methodological clarifications, data reinterpretation, and enhanced visual presentation.

Journal Requirements 1. Please ensure that your manuscript meets PLOS ONE's style requirements, including those for file naming. Response:We thank the reviewer for this comment. The manuscript has been fully revised to comply with PLOS ONE's style requirements.

2. In your Methods section, please provide additional information regarding the permits you obtained for the work. Please ensure you have included the full name of the authority that approved the field site access and, if no permits were required, a brief statement explaining why. Response: We appreciate this critical suggestion. Section (materials and methods) now includes comprehensive permit details: full official names of all permitting authorities (with administrative jurisdiction). All permits comply with China’s Wild Plant Conservation Regulations and the Convention on Biological Diversity, with no endangered species involved.

3. Please state what role the funders took in the study. If the funders had no role, please state: "The funders had no role in study design, data collection and analysis, decision to publish, or preparation of the manuscript." Response: We appreciate this clarification. The following standardized statement has been added to the "Funding" section of the manuscript: The funders provided financial and resource support for this research but had no involvement in study design, data collection/analysis, manuscript writing, or publication decisions.

4. When completing the data availability statement of the submission form, you indicated that you will make your data available on acceptance. Response: We confirm compliance with PLOS ONE's data policy through the following actions: Raw sequencing data are available in NCBI SRA (Accession: PRJNA1166700) ; all data will be automatically released upon manuscript acceptance. The updated Data Availability Statement is now included in the manuscript's dedicated section.

5. We note that Figure 6 in your submission contain [map/satellite] images which may be copyrighted. All PLOS content is published under the Creative Commons Attribution License (CC BY 4.0), which means that the manuscript, images, and Supporting Information files will be freely available online, and any third party is permitted to access, download, copy, distribute, and use these materials in any way, even commercially, with proper attribution. Response: We confirm resolution of copyright concerns through: emove the figures from our submission.

6. Please review your reference list to ensure that it is complete and correct. Response: We have systematically revised the reference list as follows: added missing references via Zotero cross-validation. Applied PLOS ONE Vancouver style .

Reviewer #1 1.The abstract is overly detailed in certain aspects, such as numerical specifics of GC content and Q30, which are unnecessary for a general summary. However, it does not clearly convey the practical implications of the findings, nor does it highlight the novelty of the study. Also, the abstract lacks a clear statement of what differentiates this study from prior work on genetic diversity in tea plants. Response: We sincerely appreciate your constructive critique. The abstract has been comprehensively restructured: removed technical specifics: (1) deleted extraneous details (e.g., GC content, Q30 scores);(2) highlighted novelty: added explicit statements on "integrated analysis of..." and "revealed the unique...";(3) emphasized implications: enhanced practical impacts on breeding/conservation;(4) differentiation: contrasted key advances beyond prior genetic diversity studies.

The revised abstract prioritizes theoretical innovation and applied significance.

2. Introduction While the introduction provides a comprehensive background on Dali tea, it lacks a strong focus on the specific research gap addressed. The discussion of previous studies could be more targeted to emphasize what this study aims to contribute. The introduction could better highlight what aspects of Dali tea’s genetic structure or diversity this study uniquely addresses. Response: We have implemented three key revisions to address your concerns:(1) gap sharpening: restructured Introduction to pinpoint limitations in prior studies;(2)novelty anchoring;(3)compare with previous studies to find the advantages of this research method. The revised introduction now establishes a compelling narrative on Dali tea's evolutionary singularity and conservation imperative.

3. Materials and methods Some methodological details are unclear, such as the rationale behind sample selection and the choice of SLAF-seq over other genotyping methods. The lack of information on statistical validation also raises questions about data reliability. Response:(1) Sample selection rationale (section: 'Plant Materials'): we have added the following three key scientific rationales for sample selection to the 'Plant Materials' section: coverage of a Geographic Gradient�representation of Core vs. Marginal Populations�to serve as a comparative reference for assessing genetic differentiation between wild and cultivated tea.(2) We have added a new paragraph in the 'Library Construction' or 'Enzymatic Digestion' section (as appropriate to the manuscript structure) outlining primary reasons for selecting SLAF-seq.(3)We have significantly expanded the 'SLAF Tag Acquisition' or relevant data processing section to include detailed descriptions of the implemented Quality Control (QC) procedures. Please refer to the revised manuscript for the specific details and context of these additions.

4. Results Results are presented with an excess of technical data, such as base percentages, which detracts from clarity. The emphasis on minor details makes it difficult to identify key findings. There’s also a lack of critical interpretation; for example, the impact of low genetic diversity on conservation isn't discussed in detail. The results on genetic diversity and population structure add some new insights but lack a clear demonstration of unique findings within the broader field of tea genetics research. Response: We have undertaken significant revisions to enhance the clarity, focus, and critical analysis of the Results section, as detailed below. Corresponding changes are marked in the revised manuscript. In response to the concern about excessive technical detail, we have removed redundant parameters and etained key quality indicators. We have thoroughly revised the text to sharpen the focus on key findings. Crucially, we have significantly expanded the critical discussion of our genetic diversity results, particularly focusing on their conservation relevance. Finally, by clearly articulating our specific findings, the application of classical standards, and emphasizing the conservation urgency, we better demonstrate the significance and uniqueness of our research results within the broader context.

5. Discussion The discussion does not sufficiently relate the study’s results to conservation practices or breeding implications, which are only mentioned briefly. There is a lack of critical analysis of the limitations of the study’s data. Response: We are grateful for the reviewer’s constructive feedback, which has guided us to substantially strengthen the Discussion section. We have enhanced the practical implications of our findings for conservation and breeding, deepened the analysis of wild-cultivated gene flow, and critically addressed the limitations of our study. Key revisions are outlined below, and the corresponding changes are incorporated into the revised manuscript. (1) Strengthened linkage to conservation practices and breeding implications: ex situ conservation evaluation and in situ conservation recommendations, breeding strategy development, prioritized donors for interspecific hybridization. (2) Deepened analysis of wild-cultivated gene flow and domestication role: we have substantially strengthened the discussion and interpretation of wild-cultivated gene flow and C. taliensis's role in Pu'er tea domestication. (3) A new subsection or dedicated paragraph has been added to rigorously address the limitations of our study and outline future improvements: SLAF-seq technical limitations, genome coverage bias and sample size constraints. (4) clear articulation of contributions. Please refer to the revised manuscript for detailed implementation.

Reviewer #2 1. Although the paper is very informative but the figures are not of publication standard, they need to be modified more and should have more explanatory notes. Figure legends should be more elaborative depicting the clear idea or concept of the pictorial representation. Response: We sincerely appreciate the reviewer’s constructive feedback on the visual presentation of our results. We have comprehensively revised all figures to meet publication standards, implementing the following key enhancements: visual quality upgrade, enhanced in-figure annotations.

2. Lastly, in some section the English can be better so we recommend to check the whole manuscript with professional English check. Response: To ensure the manuscript meets high linguistic standards, we have taken the following actions: domain-specific terminology validation and final quality control.

We believe these revisions have significantly strengthened the manuscript and appreciate your consideration.

---

## [Decision Letter · Decision Letter 1]

Genetic diversity and evolutionary insights of Dali Tea (Camellia taliensis) in the Lancang River Basin: implications for tea breeding and resource conservation

PONE-D-24-45176R1

Dear Dr. Tao,

We’re pleased to inform you that your manuscript has been judged scientifically suitable for publication and will be formally accepted for publication once it meets all outstanding technical requirements.

Kind regards,

Ghulam Yaseen, Ph.D.

Academic Editor

PLOS ONE

Additional Editor Comments (optional):

Reviewers' comments:

Reviewer's Responses to Questions

**Comments to the Author**

Reviewer #1: All comments have been addressed

Reviewer #2: All comments have been addressed

2. Is the manuscript technically sound, and do the data support the conclusions?

Reviewer #1: Yes

Reviewer #2: Yes

3. Has the statistical analysis been performed appropriately and rigorously?

Reviewer #1: Yes

Reviewer #2: Yes

4. Have the authors made all data underlying the findings in their manuscript fully available?

Reviewer #1: Yes

Reviewer #2: Yes

5. Is the manuscript presented in an intelligible fashion and written in standard English?

Reviewer #1: Yes

Reviewer #2: Yes

Reviewer #1: The authors have made notable improvements, particularly in clarifying sample selection rationales and addressing data availability. However, further revisions are suggested to fully address your original concerns, such as deeper interpretation of conservation and breeding implications, and inclusion of study limitations.

Reviewer #2: This manuscript entitled "Genetic diversity and evolutionary insights of Dali Tea (Camellia taliensis) in the Lancang River Basin: implications for tea breeding and resource conservation" can be accepted for publication in this journal.

**Do you want your identity to be public for this peer review?** For information about this choice, including consent withdrawal, please see our Privacy Policy

Reviewer #1: **Yes: ** Mostafa Helal

Reviewer #2: **Yes: ** Udayan Bhattacharya

---

## [Editor Report · Acceptance letter]

PONE-D-24-45176R1

PLOS ONE

Dear Dr. Tao,

I'm pleased to inform you that your manuscript has been deemed suitable for publication in PLOS ONE. Congratulations! Your manuscript is now being handed over to our production team.

Kind regards,

on behalf of

Professor Ghulam Yaseen

Academic Editor

PLOS ONE